# Semi-supervised atmospheric component learning in low-light image problem

**Masud An Nur Islam Fahim, Nazmus Saqib, Ho Yub Jung** * 

Department of Computer Engineering, Chosun University, Gwangju, South Korea.

* hoyub@chosun.ac.kr

**Data Availability Statement:** The data concerning this paper publicly available on https://github.com/chosun-cvlab/lowlight_semisup.

**Funding:** This study was supported by research fund from Chosun University, 2022. The funders

## Abstract

Ambient lighting conditions play a crucial role in determining the perceptual quality of images from photographic devices. In general, inadequate transmission light and undesired atmospheric conditions jointly degrade the image quality. If we know the desired ambient factors associated with the given low-light image, we can recover the enhanced image easily. Typical deep networks perform enhancement mappings without investigating the light distribution and color formulation properties. This leads to a lack of image instance-adaptive performance in practice. On the other hand, physical model-driven schemes suffer from the need for inherent decompositions and multiple objective minimizations. Moreover, the above approaches are rarely data efficient or free of postprediction tuning. Influenced by the above issues, this study presents a semisupervised training method using no-reference image quality metrics for low-light image restoration. We incorporate the classical haze distribution model to explore the physical properties of the given image to learn the effect of atmospheric components and minimize a single objective for restoration. We validate the performance of our network for six widely used low-light datasets. Experimental studies show that our proposed study achieves a competitive performance for no-reference metrics compared to current state-of-the-art methods. We also show the improved generalization performance of our proposed method which is efficient in preserving face identities in extreme low-light scenarios.

## 1 Introduction

Images captured under limited lighting conditions exhibit lower contrast, inadequate detail, and unexpected noise. Recently available photographic devices can alleviate many of the problems, but they leave artifact traces such as noise, hallows, or blurred contours. These artifacts can seriously degrade the performance of computer vision tasks apart from aesthetic issues. For example, underexposed images lead to unsatisfactory performances in tasks such as object detection, segmentation, and scene understanding [1].

Recent trends show that researchers have been focusing on developing new methods for image enhancement by blending physical models with end-to-end networks. In this context, Retinex theory [2, 3] is a well-known approach in solving conventional low-light enhancement problems. Early approaches used the handcrafted algorithm to decompose the input image into

had no role in study design, data collection and analysis, decision to publish, or preparation of the manuscript.

**Competing interests:** The authors have declared that no competing interests exist.

reflectance and illumination components, followed by a minimization step to obtain the optimal components which were then used to recover the enhanced image. The work on Retinex-net [4] made improvements by learning image decomposition adaptively for the given low-light images. However, the overall procedure has to learn the individual components and the associated optimization steps for proper reconstruction. As a result, the obtained solution often contains coarse illumination; hence, handcrafted postprocessing is required to reduce the artifacts.

Another enhancement approach [5], inspired by the hazy image recovery equation [6], directly estimates dark channels and bright channel priors for low light enhancement. However, these approaches also sometimes produce unwanted noise artifacts and improper color saturation. Even though the above approaches adopt physical models for image restoration, their underlying optimization procedures face difficult challenges involving multiple objective goals. Moreover, these methods [4, 7] utilize manual adjustments before and after restoration. Lastly, their assumptions regarding image decomposition work well under certain lighting conditions but often lack broader generalization to handle image enhancement problems.

To address the above concerns, we constructed a semi-supervised approach to learn the environmental constraints for image enhancement. We can trace the idea of integrating image dehazing theory into a low-light enhancement back to [10]. However, this earlier approach is deficient in numerous aspects, such as weak robustness and spatial fidelity. From the image haze formulation equation [6], we obtain the atmospheric light information and transmission matrix that jointly explain the haze in the natural images. We assume that low-light to hazy image transformation will enable us to incorporate the haze distribution equation. As a result, the linear inversion operation will transform to produce a "hazy image" from the given low-light image in Fig 1.

Following this, our parameter space learns the atmospheric information and transmission matrix from the inverted image and solves the dehazing equation which in turn is followed by re-inversion to produce the enhanced output. The overall restoration procedure is free of any handcrafted operations apart from image normalization. To avoid individual optimization procedures, we design a compact formulation for learning the ambient constraints altogether. In the overall training procedure, we use only 10% of the given labels for the proposed semi-supervised scheme. In essence, we propose a semi-supervised approach for learning the ambient factors from the given image through image haze distribution theory. The following summarize the significant contributions of this study:

- The proposed study uses an image quality assessment metric to regulate semi-supervised learning for low-light image restoration. This allows for only a fraction of the images having ground truths during training while achieving the desirable result.

- Our study investigates the common spatial degradation found in low-light enhancement and proposes an effective loss function combination that addresses the issues.

- Previous low-light enhancements tend to primarily rely on specific priors such as dark/bright channels, illumination, and reflection from the physical and environmental domains in order to obtain state-of-the-art results. In our study, no such priors are required to produce similar or better results.

- It is common in decomposition-based methods to rely on intensive manual postprocessing following inference. In contrast, our model can provide state-of-the-art end-to-end low-light enhancement without relying on such postprocessing.

The rest of the paper is organized as follows. Section 2 covers related works, and section 3 describes our proposed approach. In section 4, we present the detailed comparative analysis, and this is followed by the conclusion.

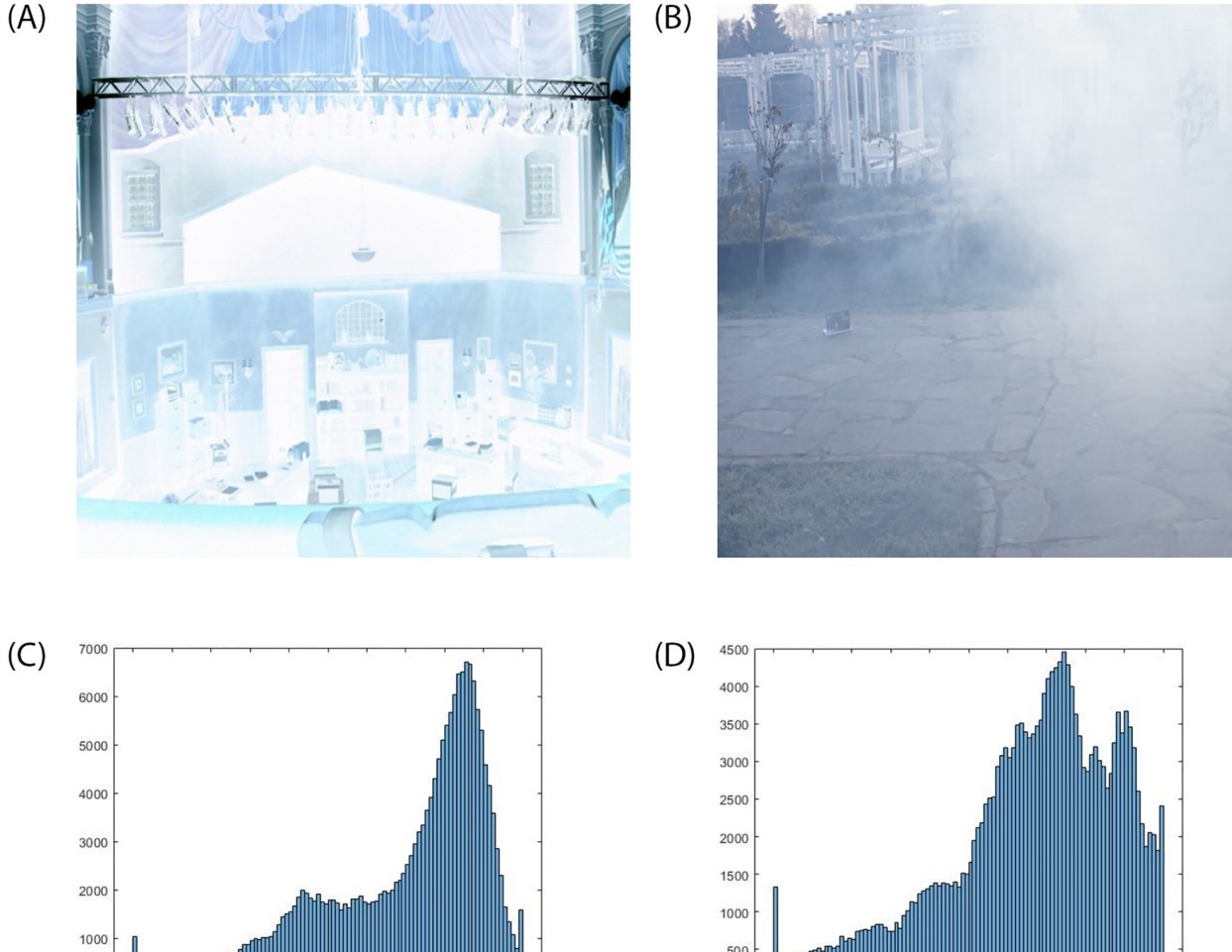

**Fig 1.** Average histogram for the 40 random hazy images from [8] dataset (1a) and similar low-light photos from the GLADNet [9] dataset (1b). After performing the inversion upon the low-light pictures, we can see similar trends in the histogram of the transformed (1c) and hazy images (1d). This statistical phenomenon influenced the proposed study to incorporate haze distribution theory for image restoration. (a) Inverse low-light image, (b) Hazy image, (c) Average histogram of inverse low light images, and (d) Average histogram of hazy images.

## 2 Related work

Image enhancement and restoration in extremely low-light conditions have been studied for a long time. This section will briefly review a line of methods that are mostly related this paper. Our method follows data driven approach and trains end-to-end neural network. However, we build physical atmospheric model, commonly used in the optimization base methods, in our network architecture to provide more complete low-light enhancement solution.

### 2.1 Optimization based methods

Traditional studies address the low-light enhancement challenge with the help of handcrafted optimization procedures or norm minimization methods. The optimization procedures, rely on assumptions, such as dark/bright channel priors [10–12]. Further, sub-optimal solutions are obtained using mathematical models such as retinex theory [4] and multi-scale retinex theory [2, 3]. Some of the studies also use handcrafted fusion input [13, 14] instead of the original image.

Earlier retinex methods [2, 3, 15, 16] use Gaussian functions to maintain dynamic range compression and color consistency. Several optimization approaches estimate the illumination map not only using adaptive [17], bilateral [18, 19], guided [20], and bright-pass [21] filters but also through derivation [10, 12], and structural assumptions [11]. Recently, to address the noise issue within retinex approaches, several methods have applied postprocessing steps such as noise fusion [22] or noise addition [23, 24]. Moreover, fusion-based approaches [5] employ background highlighting and multiple exposure information. In addition, multispectral image fusion combines the given image with an infrared image using a pseudo-fusion algorithm [25].

Despite the merits of the aforementioned methods, in general, their performance is not noise adaptive, contrast/instance aware, or artifact suppressive. Hence, they rely upon intense postprocessing, which can eventually distort minute details while increasing the computational complexity.

## 2.2 Data driven methods

Previous learning-based studies generally adopt supervised [12], semi-supervised [26], zero-shot [27], and unsupervised learning [28] for solving low light enhancement problems.

Usual supervised methods focus on solving the low-light enhancement problem through an end-to-end approach or by using theoretical schemes such as retinex decomposition. In the first category, researchers propose stable networks and customized loss functions [1, 9, 29–33], and in the second, two different objectives for reflectance and illumination are solved using novel architectures [34–36].

The study by Yang *et al.* [26] incorporates semi-supervised learning to perform image enhancement. Their work focuses on band learning from the input images, followed by decomposition and linear transformation. Zero-shot approaches focus on reducing label dependency and propose different approximation strategies. Zero-DCE [27] and Zero-DCE++ [37] obtain enhanced images by estimating multiple tone-curves from the input images. However, the computational burden is higher for these approaches than for other methods. EnlightenGAN [28] offers an unsupervised solution through an adversarial process but may lack stable generalization performance.

Our method is a hybrid of the classical atmospheric optimization approach and recent learning-based studies. As the classical methods offer an investigation of the atmospheric statistics and the machine learning studies are tuned to large datasets, bridging both will present a more capable solution, as presented in the following sections.

## 3 Atmospheric component learning

Inverse low-light images inherit very similar statistical properties to hazy natural images as seen in Fig 1. Prevalent methods [12, 38] rely on the statistical analysis of a dark primary color version of both hazy and inverted low-light images. However, these methods come with several challenges, such as multiple-variable optimization and dependency on priors that lack diverse applicability. We first observe the traditional haze distribution Eq (1) from [10], which can be integrated into our network through subsequent transformations.

$$I(x) = J(x)t(x) + A(x)(1 - t(x)). \tag{1}$$

$I(x)$ is the observed hazy image, $J(x)$ is the image we want to recover, $t(x)$ is the transmission component, and $A(x)$ is the global ambient component.

To integrate Eq (1) onto the low light enhancement problem, we first invert our low-light input image $L(x)$ and the resultant image $1 - L(x)$, which is the 'hazy image', for this problem. The hazy $I(x)$ image is replaced by $1 - L(x)$, which will be denoted as $I'(x)$. Likewise, the

recovered bright image $B(x)$ can be inverted to produce an enhanced low-light image, resulting in a simple variable replacement of Eq (1) $I'(x) = B(x)t(x) + A(x)(1 - t(x))$. If we solve for $B(x)$, we have the following.

$$B(x) = \frac{1}{t(x)}I'(x) - A(x)\frac{1}{t(x)} + A(x). \tag{2}$$

In previous works, certain assumptions were made to solve the above Eq (2). For example, earlier studies typically set transmission information to be constant over the entire environment. This assumption leads to the optimization of a single variable $A(x)$; however, it is not practical for many lighting conditions. On the other hand, prior dark channels/bright channels are extracted in some works to obtain closed-form solutions for Eq (1). The effectiveness of these practices suffer due to fluctuating transmission and ambient variables. Therefore, it is necessary to consider both to enable the best possible recovery of the given image. Accordingly, we adopt the following expression to solve over the training procedure:

$$B(x) = h(x)(I'(x) - 1) + c, \tag{3}$$

$$h(x) = \frac{\frac{1}{t(x)}\left(I'(x) - A(x)\right) + (A(x) - c)}{I'(x) - 1}. \tag{4}$$

We have taken some algebraic liberties to formulate $h(x)$ in Eq (3). With $h(x)$ formulated as in Eq (4), the hazy Eq (2) can be approximated to Eq (3), where $c$ is constant and $h(x)$ is the atmospheric component that combines the input image, transmission information, and ambient information. We set $c = 1$ to transform the negative result from the left part of the equation into a bounded positive value.

The specifics of neural network architecture are not important in our approach. As we aim to learn the compact representation of the Eq (4), we can use any stable network to learn $A(x)$ and $t(x)$ to find $h(x)$ with cited equation. See Fig 2. Afterward, by simply plugging to the Eq (3), we can obtain our desired enhanced image. In this paper, we used DnCNN [39] to learn $A(x)$ and $t(x)$. Table 1 provides the necessary layer specifications of the DnCNN network.

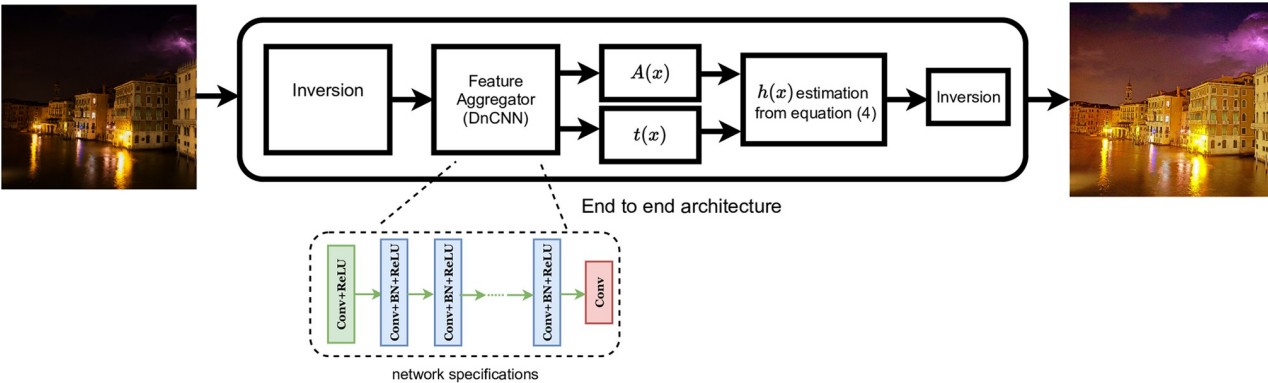

**Fig 2. Schematic network of the proposed study.** We perform image inversion in the beginning and end to simulate hazy image. We use DnCNN [39] as the feature aggregator, which internal architectural details are provided here. The network approximates the atmospheric components of Eq (4) with semi-supervised training and the reformulated hazy Eq (3) is solved before the inversion.

**Table 1. The layers for DnCNN model.**

| Layer name | Layer | Network parameter |
|---|---|---|
| Feature extraction | Conv+ReLU | 3 3,64 |
| Feature conversion | Conv+BN+ReLU | 3 3,64 |
| Residual image generation | Conv | 3 3,64 |

In the following subsection, we propose semi-supervised learning for approximating $h(x)$. Eq (3) is solved using non-trainable layers inside the semi-supervised trained network before sending $B(x)$ to the loss layer (see Fig 2). In this way, we can learn the effect of the ambient factors on the given image.

### 3.1 Validation metric for semi-supervised learning

The proposed semi-supervised learning starts with a few hundred images with ground truths. After the initial training, new images without ground truths can be added to the training set by using the results from the initially trained network as ground truths. Naively, it may seem that the new training images will simply reinforce the current state of the network. However, this is avoided when we use a multi-objective loss function with smoothness and brightness costs so that a network response that is the same as the ground truth does not necessarily produce the minimum loss. Likewise, we can choose only the images that produce an 'objectively correct' network response. The correctness of the network response image can be validated by image quality assessment metrics that are not based on the ground-truth reference image.

For this study, we used the GLADNet dataset [9] for training and the Naturalness Image Quality Evaluator(NIQE) [40] metric to validate the correctness of the network response. Among no-reference image quality assessments (IQA) like PIQE (Perception Image Quality Evaluator) [41], NIQE (Naturalness Image Quality Evaluator) [40] or BRISQUE (Blind/Referenceless Image Spatial Quality Evaluator) [42], NIQE is the most widely accepted metrics which has been used in many previous studies [1, 27, 34] for quantifying the deviation from image quality. There are more recent no-reference metrics like UNIQUE [43], however more well known NIQE metric is chosen as the validation metric. Simultaneously, among several highly recognized low-light datasets, only GLADNet and LOL [37] preserve corresponding ground truths. However, LOL contains only the indoor scenes, where as most of the low light images are from outdoors. Therefore, we use the GLADNet dataset as the training data for our proposed approach.

Firstly, we adopt 500 random images from the GLADNet dataset with their corresponding ground truths. The NIQE scores of the 500 ground-truth images were then computed. Since these images have proper light distribution, their NIQE scores are much lower than their low light counterparts. The average of the precomputed NIQE scores of the preselected reference image is denoted as $N_a$. The network was trained with this small subset until primary local minima are reached for the network.

$R_1$ denotes the large set of images without ground-truth images. The network response images from $R_1$ are computed as $\hat{R}_1$, and NIQE scores for $\hat{R}_1$ images are calculated. To select the subsequent images to be included for retraining, we only include images that have response images with *NIQE* scores that are very near the precomputed value $N_a$. Thus, the results from unlabeled images must be validated by the desired NIQE score before inclusion in the training set.

If $m$ images from $\hat{R}_1$ pass the NIQE metric validation, the training set will be updated with $500 + m$ paired images, of which 500 will have true ground truth and $m$ will have images from $\hat{R}_1$ as acting ground truths. In this way, we start the retraining with our mixed labels and repeat until we update the parameter space with the whole GLADNet dataset of 5000 samples. We experimented with different starting subsets from the GLADNet dataset but did not observe any significant deviance from the reported result. Up to 5 rounds of retraining was required to cover the entire dataset with high NIQE scores.

Because the new training images with acting ground truths are validated with the NIQE metric, the resultant network produces desirable NIQE scoring images as a side effect. The target loss function itself does not include the NIQE score in its formulation, which allows NIQE to be a viable validation metric. The specifics of the loss function are introduced in the next subsection.

### 3.2 Loss function

Inferences from typical end-to-end models may contain artifacts such as over-smoothing, lack of contrast, or traces of convolution operations. Hence, some approaches rely on handcrafted post-inference tuning as an extension of the method. To avoid such manual operations, we propose a combined loss function with the aims of reducing blurry edges, suppressing noise, and producing adaptive contrast independent of the domain while achieving high image quality metric scores.

We experimented with different loss function setups, and the following loss function is proposed.

$$L_{total} = \lambda_1 L_1 + \lambda_2 L_{brightness} + \lambda_3 L_{smooth} + L_{SSIM}. \tag{6}$$

We started with the $L_1$ loss function, which is the distance between the ground truth and the prediction image. To go further, we propose $L_{brightness}$ loss to introduce more light information during prediction. The equations for the $L_1$ and $L_{brightness}$ are as follows:

$$L_1 = \frac{1}{n}\sum_{j=1}^{n}|y_g - y_p|, \tag{6}$$

$$L_{brightness} = \frac{1}{n}\sum_{j=1}^{n}|y_g^{\gamma_1} - y_p^{\gamma_2}|. \tag{7}$$

$y_g$ is the ground truth, and $y_p$ is the prediction, which is universal for both supervised pre-training and semi-supervised iterations. For the $L_{brightness}$ loss function, $y_g^{\gamma_1}$ and $y_p^{\gamma_2}$ are the gamma-corrected ground truths and the predictions respectively. Our $L_{brightness}$ loss function first darkens the predicted image and brightens the corresponding labels through gamma correction and then measures the $L_1$ distance between them.

The $L_{smooth}$ loss measures the distance between the prediction and the corresponding smoothed label image by median, Gaussian, consecutive upsampling and downsampling.

$$L_{smooth} = \frac{1}{n}\sum_{j=1}^{n}|y_g^{smooth} - y_p|. \tag{8}$$

If $L_{smooth}$ is the only loss function for the overall training process, it will constrain the parameter space to infer a smoothed-out enhanced image. In many of the previous methods, smoothing was obtained through postprocessing in order to reduce the remaining noise. Instead, we crafted the loss function to learn smoothing during training.

The final component is the SSIM loss, which constraints the parameter space to adapt to image metric performance.

$$L_{SSIM} = \frac{1}{n} \sum_{j=1}^{n} 1 - SSIM(y_g, y_p). \tag{9}$$

The $L_1$ loss, $L_{brightness}$ loss for brighter prediction, noise-suppressing $L_{smooth}$, and the $L_{SSIM}$ loss together aid in achieving overall perceptual and structural fidelity. To successfully inject the influences of all the loss functions into the architecture, we empirically tuned the parameters to $\lambda_1 = 0.35$, $\lambda_2 = 0.5$, and $\lambda_3 = 0.15$. For the gamma correction, $\gamma_1 = 0.85$, and $\gamma_2 = 1.15$ were chosen through the convex sum. Fig 3 shows the results of an identically initialized network with individual loss functions $L_{total}$, $L_1$, $L_2$, $L_{brightness}$, $L_{smooth}$, and $L_{SSIM}$ and the proposed combined loss function. The efficacy of the proposed loss function in comparison to each loss function is shown.

Fig 4 represents the visual effect of loss function weights. The figure shows the impact of a single parameter awhile keeping the default values for other parameters. $\lambda$ and $\gamma$ parameters are empirically fitted using training data. The best combination of parameters were estimated according to several experiments on the training dataset. Without balanced combination between the parameters, the network's results can deviate with hallow effect, exaggerated details, undesired spatial smoothness, lower contrast, and unnatural color presence.

## 4 Experiments

In this section, we demonstrate the performance of the proposed method in comparison to eleven contemporary studies. In the following, we present the experimental settings and then show the qualitative and quantitative evaluation on six widely used datasets over three different metrics.

### 4.1 Experimental setup

In our training process, we use the Adam optimizer with a learning rate of 0.0001. We utilize the learning rate decay from the original TensorFlow library, where we monitor the validation SSIM to decay the learning rate. Over the total training time, we used a batch size of 16 and normalized all images between 0 and 1. We adopt the usual data augmentation procedure for the overall training procedure. In our training setup, we did not fragment the training images to smaller patches. Additionally, we keep the image sizes to be their original size during the training process. We adopt the training dataset from the GLADNet dataset. This dataset contains 5000 low-light images with their corresponding labels. The images in the dataset are not limited to any specific environment or class instances and contain both indoor and outdoor photos taken in daylight or nighttime of humans, animals, and natural images.

### 4.2 Comparison

This section will provide a short description of the datasets we have used, the compared methods, and their performances.

**Datasets.** We apply our testing procedure to six datasets: LIME [16], LOL [37], MEF [49], NPE [21], DICM [48], and VV [47].

**Compared studies.** For comparison, we use the following methods: Dong et al. [10], LIME [16], CRM [44], fusion [24], semi-decoupled [45], MBLLEN [1], KinDL [34], KinDL++ [36], EnlightenGAN [28], DeepUPE [46], and Zero-DCE [27]. Since most test images do not come with corresponding ground truths, we use no-reference metrics such as PIQE [41], NIQE [40], and BRISQUE [42]. NIQE signifies the deviations of statistical regularities for a given image

(A)

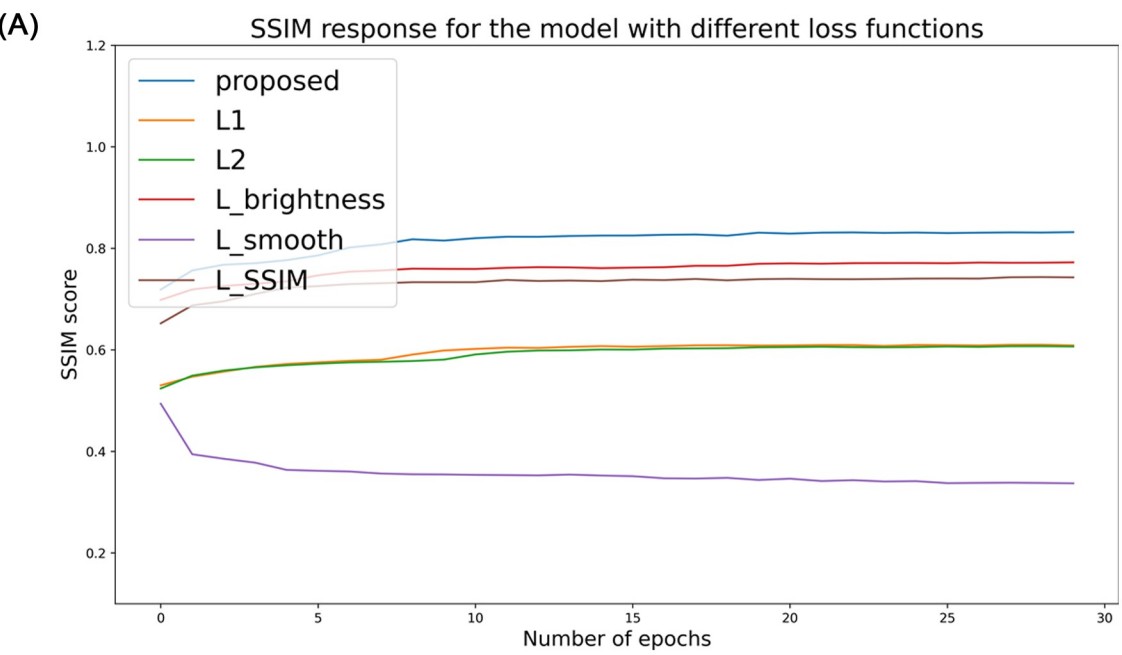

(B)

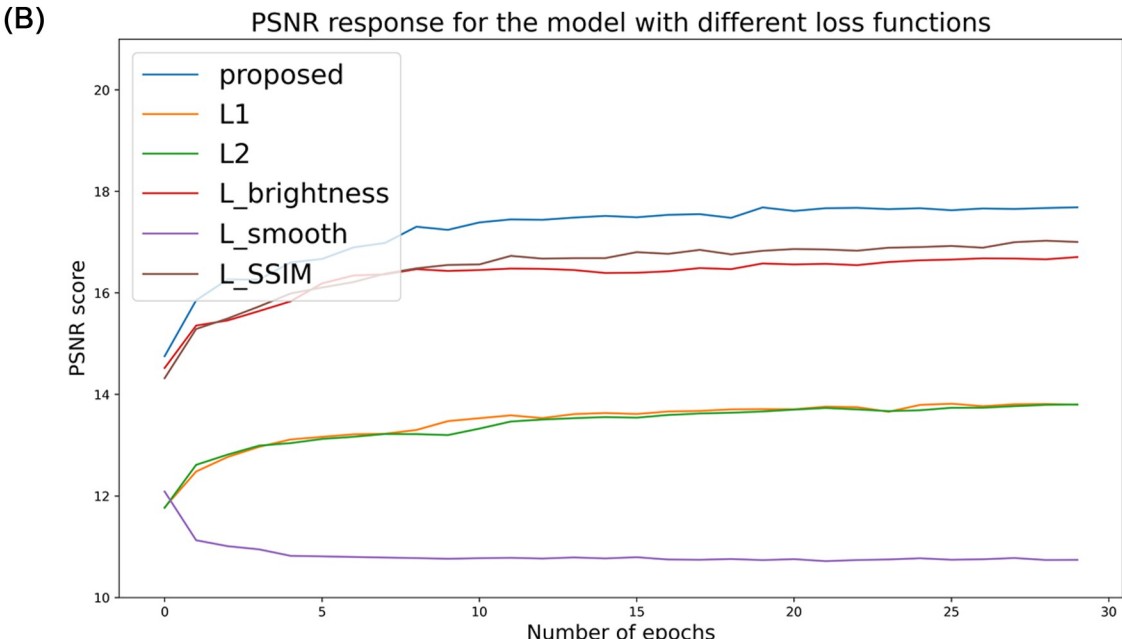

**Fig 3. The response for the each loss functions used in our work.** Under the same initialization, proposed combined loss-function avails better SSIM and PSNR response than the individual loss functions. (a) Epoch vs. SSIM response due to the loss functions and (b) Epoch vs. PSNR response due to the loss functions.

without any reference. Similarly, PIQE inversely corresponds to the perceptual quality of the given image. BRISQUE also operates blindly and utilizes locally normalized luminance coefficients to compute the image quality deviation. Thus, lower score is better for all three metrics. Tables 2–4 show that our method achieves state-of-the-art or comparable performances for the given datasets and produces the best average score across all datasets.

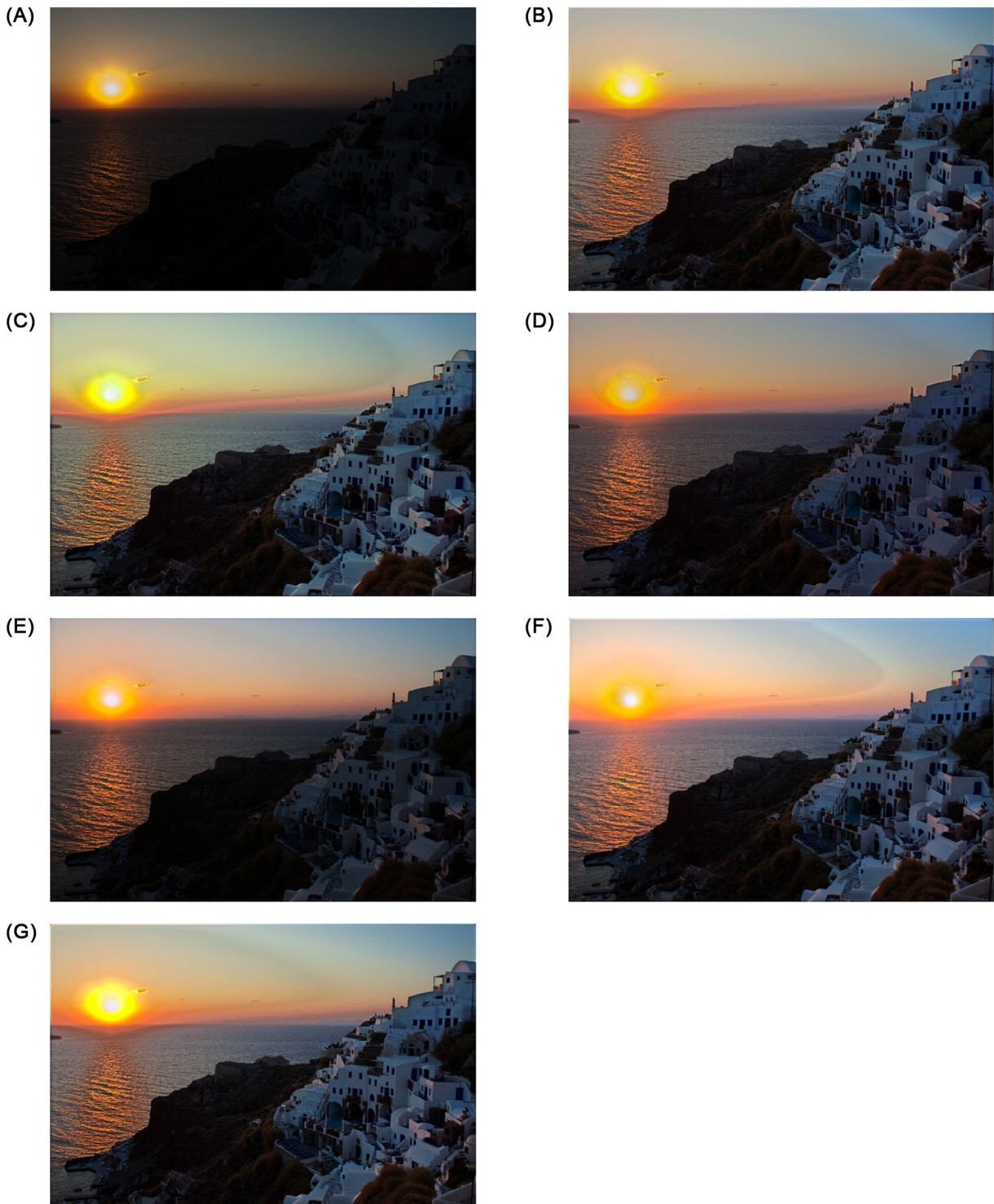

**Fig 4. The above figure shows the effect of the loss function parameters.** $\lambda$s are weights for $L_1$, $L_{brightness}$, and $L_{smooth}$. $\gamma$s are parameters for $L_{brightness}$. Without the balanced parameters, the network produces deviated outputs. (a) Input Low-light image, (b) Proposed, (c) $\lambda_1 = 0.05$, (d) $\gamma_1 = 0.2$, (e) $\gamma_2 = 2$, (f) $\lambda_2 = 1$, and (g) $\lambda_3 = 0.85$.

**Table 2. Perception Image Quality Evaluator (PIQE) comparison between nine low light enhancement methods on six different datasets.** Best score is in bold and second best is underlined.

| Method/Dataset | VV [47] | DICM [48] | LIME [16] | LOL [37] | MEF [49] | NPE [21] | Average |
|---|---|---|---|---|---|---|---|
| LIME [16] | 19.8520 | 16.3735 | 12.6429 | 13.6715 | 16.8422 | 13.7836 | 15.5276 |
| Dong et al. [10] | 13.2391 | 13.9523 | 14.5912 | 19.1315 | 18.4940 | 14.4011 | 15.6348 |
| MBLLEN [1] | 11.5750 | 16.1056 | 16.1573 | 13.1625 | 20.3220 | 13.6689 | 15.1652 |
| CRM [44] | 12.2987 | 15.6655 | 13.1111 | 19.1352 | 18.4965 | 13.1257 | 15.3054 |
| Fusion [24] | 9.0956 | 15.6556 | 12.1613 | 17.8952 | 19.9009 | 13.5352 | 14.7073 |
| KinDL [34] | 10.7151 | 11.9790 | 12.8772 | 9.3091 | 11.6322 | 16.3012 | 12.1356 |
| KinDL++ [36] | 10.7045 | 7.9980 | 13.5777 | 8.7117 | 9.5811 | 16.1812 | 11.1257 |
| Semi-Decoupled [45] | 9.3511 | 17.587 | 12.5561 | 27.7175 | 22.6213 | 16.5653 | 11.7347 |
| Zero-DCE [27] | 9.6766 | 8.1825 | 9.6177 | 11.3682 | 7.6688 | 10.9502 | 9.5773 |
| EnlightenGAN [28] | 9.9112 | 12.2223 | 13.1718 | 11.9347 | 15.4127 | 13.0161 | 12.6030 |
| DeepUPE [47] | 11.3748 | 9.6465 | 14.2783 | 13.0920 | 17.3246 | 17.0602 | 13.8321 |
| Proposed | **7.0877** | **7.5725** | 10.0965 | 11.2002 | **7.0011** | 11.0311 | **7.8169** |

**Table 3. Naturalness Image Quality Evaluator (NIQE) comparison between eleven low light enhancement methods on six different datasets.** The best score is in bold and the second best is underlined.

| Method/Dataset | VV [47] | DICM [48] | LIME [16] | LOL [37] | MEF [49] | NPE [21] | Average |
|---|---|---|---|---|---|---|---|
| LIME [16] | 3.9788 | 3.5009 | 4.9526 | 4.1368 | 4.1155 | 3.4458 | 4.0217 |
| Dong et al. [10] | 4.2628 | 4.2002 | 4.2021 | 3.8971 | 4.6315 | 3.8263 | 4.1700 |
| MBLLEN [1] | 3.5721 | 4.9279 | 4.3491 | 3.9303 | 3.2391 | **3.3416** | 3.8933 |
| CRM [44] | 3.7216 | 4.2148 | 4.3552 | 3.1583 | 4.7578 | 4.3230 | 4.0884 |
| Fusion [24] | 3.8535 | **3.3375** | **3.4659** | 3.2639 | 3.9106 | 3.3991 | 3.5384 |
| KinDL [34] | 3.4269 | 3.5401 | 4.3527 | **3.0722** | **2.7614** | 3.9581 | 3.5185 |
| KinDL++ [36] | 3.5726 | 3.9136 | 4.1922 | 3.8945 | 3.2907 | 4.3072 | 3.8618 |
| Semi-Decoupled [45] | 4.8923 | 4.1358 | 5.3796 | 5.8426 | 5.5946 | 4.2899 | 6.5266 |
| Zero-DCE [27] | 10.504 | 11.7541 | 14.1206 | 7.3112 | 12.1316 | 10.7102 | 12.8393 |
| EnlightenGAN [28] | 3.9012 | 3.8521 | 4.4123 | 3.9751 | 4.1725 | 3.7451 | 5.9732 |
| DeepUPE [46] | 4.2258 | 3.9651 | 5.0751 | 4.6723 | 3.8122 | 4.0943 | 4.3052 |
| Proposed | **2.7642** | 3.4239 | 3.7071 | 3.3188 | 3.0102 | 3.8009 | **3.3375** |

**Table 4. Blind/Referenceless Image Spatial Quality Evaluator (BRISQUE) comparsion between nine low light enhancement methods on six different datasets.** Best score is in bold and second best is underlined.

| Method/Dataset | VV [47] | DICM [48] | LIME [16] | LOL [37] | MEF [49] | NPE [21] | Average |
|---|---|---|---|---|---|---|---|
| LIME [16] | 27.5223 | 36.9602 | 25.8193 | 25.8509 | 38.9459 | 21.9541 | 29.5087 |
| Dong et al. [10] | 28.5563 | 45.5648 | 27.0681 | 27.9744 | 40.6047 | 26.6699 | 32.7413 |
| MBLLEN [1] | 24.0018 | 25.4718 | 27.4972 | **13.8857** | 27.6337 | 20.8256 | 40.1556 |
| CRM [44] | 27.0305 | 34.8725 | 26.8219 | 21.5269 | 38.0785 | 23.0998 | 28.5716 |
| Fusion [24] | 25.8686 | 35.9418 | 26.9049 | 19.9002 | 38.7504 | 22.8034 | 28.3615 |
| KinDL [34] | 23.1707 | 32.4027 | 32.3042 | 26.9681 | 33.3069 | 20.1546 | 28.0512 |
| KinDL++ [36] | 22.8755 | 31.1161 | 37.4435 | 22.2963 | 32.9189 | 20.4419 | 27.8487 |
| Semi-Decoupled [45] | 24.2539 | 33.7456 | 26.6086 | 38.4748 | 35.5634 | 22.3599 | 30.1677 |
| Zero-DCE [27] | 34.2122 | 23.6767 | 22.0031 | 32.0853 | 25.2623 | 25.6566 | 27.1694 |
| EnlightenGAN [28] | 25.1723 | 34.9652 | 27.4341 | 30.6912 | 21.4321 | 26.8712 | 27.7651 |
| DeepUPE [46] | 28.3457 | 36.2712 | 26.1122 | 23.1612 | 23.7512 | 31.4325 | 28.6123 |
| Proposed | **22.4367** | **17.0272** | **19.0254** | 32.2238 | **13.3232** | **15.3297** | **21.5610** |

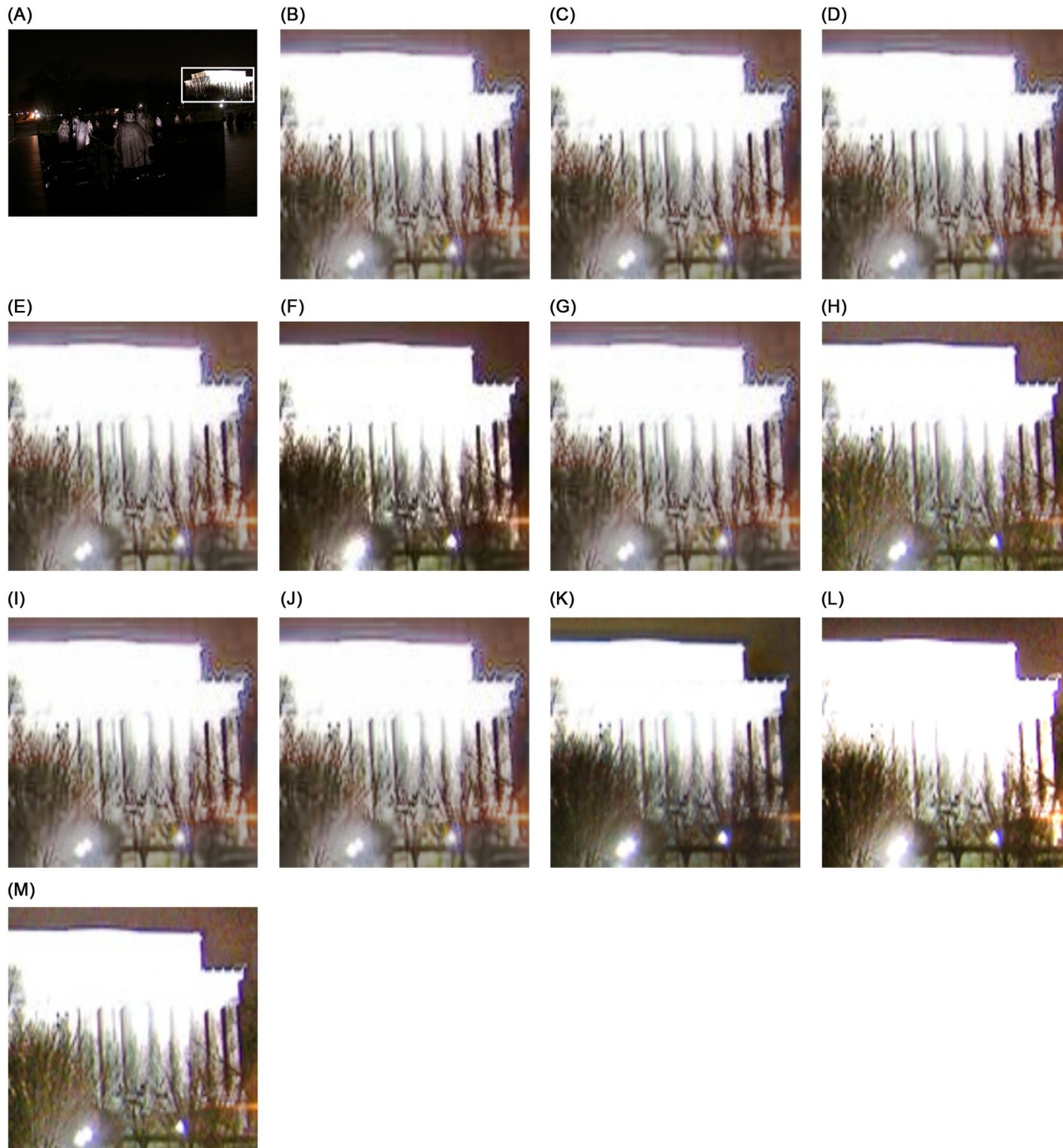

**Fig 5. This figure compares the night-time image reconstruction performance.** The figure shows the presence of noise and the trace of convolution in many of the restored images. In comparison to other methods, the proposed approach can recover the scene without perturbing the homogeneity of the foreground and background (for best view, zoom-in is recommended).

From Figs 5 and 6, we can see the restoration difference between the proposed study and the compared studies. The studies used for comparison show traces of convolution on the edge line, noisy reconstruction, color distortion, and overly contrasted stretching in the restored image. We have also highlighted the specific portion for all methods to demonstrate

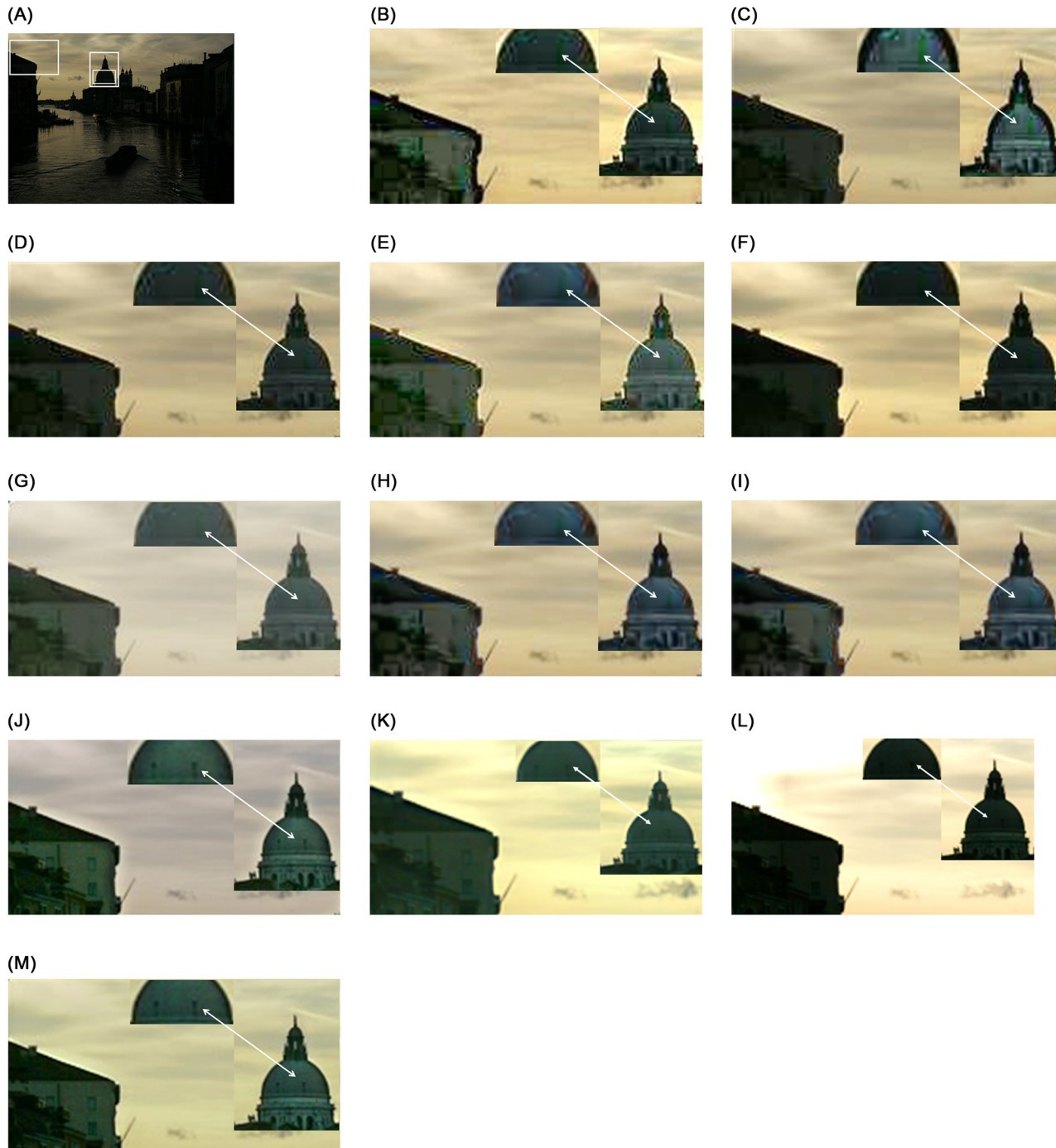

**Fig 6. This figure shows the typical low light image enhancement comparison.** We highlight the window recovery for the dome. The propose approach is one of the few methods that can recover the windows while achieving desired lighting condition. Additionally, we see proposed study can preserve the ambient contrast by suppressing the over-whitening (for best view, zoom-in is recommended).

the restoration complexities. For example, in Fig 6, only our approach and Zero-DCE [27] can successfully recover the cathedral's dome with its windows, while other methods leave it without any window and show noisy restoration. These visual results demonstrate that the proposed method can restore low-light images with minute details while preserving the

naturalness of the content while exhibiting as much noise-suppressive behavior as possible. Additionally, better and more competitive metric performances show our method's effectiveness in diverse datasets. The our comprehensive dataset results are found in https://github.com/chosun-cvlab/lowlight_semisup.

### 4.3 Identity preservation for image restoration

Low-light image enhancement is vital and mandatory postprocessing in smartphone cameras to ensure visual clarity and aesthetics. Recent trends show that low-light camera performance is an acid test for smartphone cameras. The proposed network was able to suppress noise, preserve details, restore instance-aware colors, and maintain overall fidelity. We have validated our network through both qualitative visual performance and quantitative evaluations for six popular datasets.

However, one major issue with deep-learning studies is dataset bias. Due to this, some studies show gender bias during the inference stage. In Fig 7, KinDL [34], and KinDL++ [36] identify the woman in the low-light image but also infer lipstick on the enhanced image of the woman, which contradicts the low-light face without lipstick mark. As the unsupervised methods promise to be unbiased, Zero-DCE [27] and EnlightenGAN [28] studies show no such reconstruction. Our network also preserves the woman's actual identity and does not show any assertion bias for the woman, even with semi-supervised training.

Furthermore, in the same figure, our network also captures the background people and preserves the human face intact during inference. In comparison, the compared methods interpolate the background people's faces as blobs, which can significantly damage person recognition performance. In the third row of the Fig 7, our network can successfully infer the human face without any out-of-domain artifacts or damaging the person's identity. These minute performance gains can significantly affect the efficacy of surveillance systems or general person identification-related applications.

In the end, it is difficult to quantify a network's limitations on these biases; how many different biases are present, is the data set or the network more liable for the bias, or what is the guarantee that it will be bias free in real-life scenarios? Despite these challenges, our study presents a lightweight network with only 11k parameters and shows significant improvements through semi-supervised learning. At least for these examples, it was able to maintain person's identity and did not assert gender bias to achieve instance-aware low-light enhancement performances.

## 5 Conclusion

This study aims to deliver an effective solution for low-light enhancement by learning the atmospheric components from the given image. Primarily, atmospheric elements allow us to enhance low-light images with minimal artifacts. Furthermore, we address the low-light image enhancement through the semi-supervised approach to reduce the need for ground-truth paired images and dataset bias.We have used a simple deep convolution network to estimate atmospheric components. However, since our method combines physical models with deep learning approaches, our contribution is fundamentally independent of the network design. We can use any of the established network architectures to extract the compact representation for Eq (3). Like a plug-n-play method, any network such as U-net can be used to approximate the compact form of the $h(x)$ in Eq (4), followed by direct plugging to Eq (4) to solve the inverse problem. To establish the overall scheme, we propose a combined loss function that applies the necessary constraint to enable the network to learn sophisticated features from the training domain.

Our experimental results show that the proposed scheme can restore low-light images with high fidelity and achieve satisfactory performance in six different datasets compared to eleven

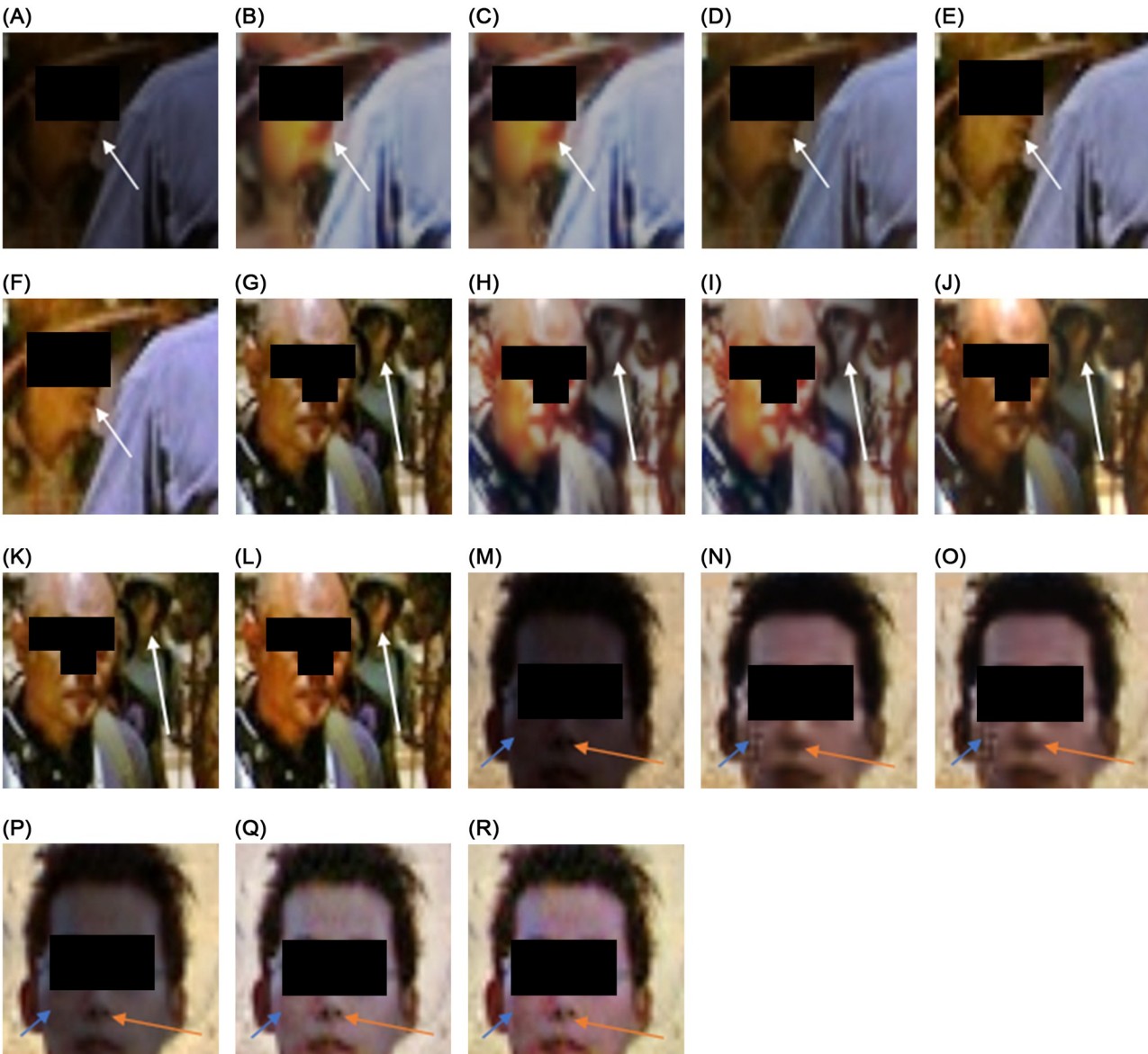

**Fig 7. Demonstration of identity preservation.** The previous data-driven methods assign the lipstick marking on the woman, but lipstick is not present in the original images. The proposed approach is free from such bias, even though it was semi-supervised. In the second row, the proposed method can recover the background faces context-coherently, whereas other data-driven methods tend to blend the faces with the environment light. The proposed approach in the last row faithfully reconstructs the person's nose as with other unsupervised methods. Again, the previous data-driven method's noses differ significantly from the low-light face (for best view, zoom-in is recommended).

state-of-the-art studies. For three different no-reference metrics, our method experiences lowest performance drops compared to the existing methods over the six different datasets.

We also discuss our concerns about the biases influencing the deep network's performance during image restoration. In the future, we aim to focus on more elegant learning approaches and to increase the generalization ability for image restorations while augmenting their performances in diverse vision applications. Additionally, we plan to work on analyzing biases within deep computational photography applications.

## Author Contributions

**Conceptualization:** Masud An Nur Islam Fahim.

**Data curation:** Nazmus Saqib.

**Formal analysis:** Masud An Nur Islam Fahim.

**Funding acquisition:** Ho Yub Jung.

**Methodology:** Masud An Nur Islam Fahim.

**Project administration:** Ho Yub Jung.

**Resources:** Ho Yub Jung.

**Software:** Masud An Nur Islam Fahim.

**Supervision:** Ho Yub Jung.

**Validation:** Nazmus Saqib.

**Visualization:** Nazmus Saqib.

**Writing – original draft:** Masud An Nur Islam Fahim.

**Writing – review & editing:** Nazmus Saqib, Ho Yub Jung.

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
