## [Decision Letter · Decision Letter 0]

17 Jan 2023

PONE-D-23-00220Semi-supervised atmospheric component learning in low-light image problemPLOS ONE

Dear Dr. Jung,

Thank you for submitting your manuscript to PLOS ONE. After careful consideration, we feel that it has merit but does not fully meet PLOS ONE’s publication criteria as it currently stands. Therefore, we invite you to submit a revised version of the manuscript that addresses the points raised during the review process.

We look forward to receiving your revised manuscript.

Kind regards,

Praveen Kumar Donta, Ph.D.

Academic Editor

PLOS ONE

Journal Requirements:

"This study was supported by research fund from Chosun University, 2022."

"This study was supported by research fund from Chosun University, 2022. The funders had no role in study design, data collection and analysis, decision to publish, or preparation of the manuscript."

5. We note that Figure 7 includes an image of a [patient / participant / in the study]. 

Reviewers' comments:

Reviewer's Responses to Questions

**Comments to the Author**

1. Is the manuscript technically sound, and do the data support the conclusions?

Reviewer #1: Yes

Reviewer #2: Yes

2. Has the statistical analysis been performed appropriately and rigorously? 

Reviewer #1: Yes

Reviewer #2: Yes

3. Have the authors made all data underlying the findings in their manuscript fully available?

Reviewer #1: No

Reviewer #2: Yes

4. Is the manuscript presented in an intelligible fashion and written in standard English?

Reviewer #1: Yes

Reviewer #2: Yes

5. Review Comments to the Author

Reviewer #1: In this work, a hybrid of the classical atmospheric optimization approach and recent learning-based studies is proposed to address the low-light image enhancement problem, which is expected to bridge the benefits of two genres of techniques.

1, It should be explicitly noted in the manuscript that a lower NIQE score indicates a higher perceptual image quality.

2, This paper randomly select 500 images from the GLADNet dataset to start the model training, and then adopt the proposed semi-supervised learning scheme to train the model. A natural question is that why not direclty use all 5000 images with ground-truth to train the model using standard supervised learning? What's the advatange of the proposed semi-supervised learning over the supervised learning?

3, A mixed loss function is adopted to train the model. There should an ablation study to verify the contribution of each single loss term.

4, In addition to the adopted performance measure, the authors are recommended to also use UNIQUE[1], a SOTA NR-IQA metric, to validate the performance of competting methods.

[1] Zhang, W., Ma, K., Zhai, G., & Yang, X. (2021). Uncertainty-aware blind image quality assessment in the laboratory and wild. IEEE Transactions on Image Processing, 30, 3474-3486.

Reviewer #2: This paper proposes an end-to-end low-light enhancement network model based on the fact that the inverse and fogged images of low-light images have similar statistical information, and a semi-supervised learning strategy for tuning low-light image enhancement using the no-reference image quality evaluation metric (NIQE). The paper presents qualitative and quantitative comparisons with current mainstream methods on several common publicly available low-light datasets.

Strengths

1. The proposed semi-supervised training strategy is relatively novel and the obtained results also achieve optimal and suboptimal performance on multiple datasets.

2. The trained network model produces smaller inference bias compared to other methods which is demonstrated in Section 4.3.

Weaknesses

1. The specific structure of the network model is not described in detail. The physical model is from previous work LIME, which is slightly less innovative on network model structure.

2. Is it problematic to use the NIQE metric as a quantitative metric for comparison with other methods while the training strategy already uses it? Despite the small amount of explanation provided by the authors in the paper, more discussion on the rationality of the approach is still needed.

6. PLOS authors have the option to publish the peer review history of their article (what does this mean?). If published, this will include your full peer review and any attached files.

Reviewer #1: No

Reviewer #2: No

---

## [Author Response · Author response to Decision Letter 0]

9 Feb 2023

We have the same response to reviews in pdf format in the submission package. The pdf format is stylistically organized better, and easier to read.

We thank the reviewers for their kind critiques. We hope we were able to answer all of the concerns to satisfaction. We also want the reviewers to know that we will make all the codes and image results available upon the publication of the paper.

Reviewer 1: It should be explicitly noted in the manuscript that a lower NIQE score indicates better perceptual quality.

Response: In 3.1, we added that NIQUE quantifies the deviation from the image quality. We hope we made the paper more clear. We also noted in the evaluation section that a lower value is better for all three evaluation metrics.

Reviewer 1: This paper randomly selects 500 images from the GLADNet dataset to start the model training and then adopt the proposed semi-supervised learning scheme to train the model. The natural question is why not directly use all 5000 images with ground truth to train the model using standard supervised learning? What's the advantage of the proposed semi-supervised learning over supervised learning?

Response: We agree and understand that using more data with ground-truth will almost always produce a better result. The advantage of semi-supervised learning is that it requires fewer number of images with ground-truths compared to direct supervised learning. And ground-truths are not always easy to obtain. Thus, we hoped that one of the contribution is to show that even with smaller number of ground-truths, we can produce state-of-the-arts results. Essentially, however, the difference between using 500 and 5000 ground-truths were sufficiently small enough to give a merit to semi-supervised learning, where we can make an argument for an additional contribution.

Reviewer 1: A mixed loss function is adopted to train the model. There should be an ablation study to verify the contribution of each single loss term.

Response: Fig. 3 shows differences of SSIM and PSNR for each single loss term as they are being trained. Instead of using no-reference metrics, we did our ablation test using SSIM and PSNR from ground-truths. Interestingly enough, a single SSIM loss does not produce highest SSIM score in the ablation test images. This is probably due to lack of sufficient diversity or number of training data, which our approach can compensate through the physical model and mixture of different loss functions such as smoothness and brightness functions. 

Fig. 4 also shows qualitative assessment of different weight values in the mixed loss functions. We understand that more can be done to make our ablation study more complete, however, we hope the study presented here are of sufficient interest for publication.

Reviewer 1: In addition to the adopted performance measure, the authors are recommended to also use UNIQUE [Zhang et al. 2021], a SOTA NR-IQA metric, to validate the performance of competing methods.

Response: Thank you for referring to UNIQUE [Zhang et al. 2021], a good study on blind image quality assessment. In our study, we have cited this paper. At present, unfortunately, we are not able to include a full-fledged comparison of the UNIQUE metric for all of the datasets and compared studies. We have observed UNIQUE's wide acceptance in the BIQA studies, but for image restoration studies, this metric is comparatively new to the community. In our future study, we will investigate the prominence of the UNIQUE metric over restorations task from different domains and aim to present a versatile assessment. In the meantime, we hope our current manuscript with three different blind evaluation metrics is sufficient to present the merit of our study.

Reviewer 2: The specific structure of the network model is not described in detail. The physical model is from previous work LIME [Guo et al. 2017], which is slightly less innovative on network model structure.

Response: We use DnCNN [Zhang et al. 2017] for our network. We have updated the Fig.2 and the manuscript to describe the model. However, unlike many of the previous studies, we want to emphasize that structure of the network is not important to the proposed approach. In Fig. 2, our feature map aggregator is an universal block, which can be any network with a slight modification (One input, two outputs instead of one). Hence, any one of networks like ResNet/DnCNN/UNet [He et al 2016, Zhang et al. 2017, Ronneberger et al. 2015], etc. can be used as the core feature extractor, which learns the approximate features for $A(x)$ and $t(x)$ to solve equation (4) for low-light enhancement.

Reviewer 2: Is it problematic to use the NIQE metric as a quantitative metric for comparison with other methods while the training strategy already uses it? Despite the small amount of explanation provided by the authors in the paper, more discussion on the rationality of the approach is still needed.

Response: We understand the reviewer's concern, however, it is common for image restoration research to use the same metric for both training and testing. For example, typical super-resolution networks use SSIM/PSNR as the regularizer/loss function and the main comparison metric [Galindo and Pedrin 2019]. 

Additionally, our method does not provide network gradients to the neural network with respect to NIQE. NIQE merely acts as a selector for the next training batch. Indeed, an implicit bias from NIQE may be present, but the bias should not be great as methods that provide training gradient directly from test/training metric. 

We are aware of the fidelity concern for the no-reference metrics, especially NIQE, and yet, among regular no-reference metrics, we have observed NIQE is the most used by the researchers [Banik et al 2018, Zhang et al. 2022]. Therefore, by following our peers, we have used NIQE as a selector during the training and one of the validation metrics. We added more details on why we choose NIQE in section 3.1.

References

Partha Pratim Banik, Rappy Saha, and Ki-Doo Kim. Contrast enhancement of low-light image using histogram equalization and illumination adjustment. In 2018 International Conference on Electronics, Information, and Communication (ICEIC), pages 1–4, 2018. 

Eldrey Galindo and Helio Pedrini. Image super-resolution improved by edge information. In 2019 IEEE International Conference on Systems, Man and Cybernetics (SMC), pages 3383–3389, 2019. 

Xiaojie Guo, Yu Li, and Haibin Ling. Lime: Low-light image enhancement via illumination map estimation. IEEE Transactions on Image Processing, 26(2):982–993, 2017. 

Kaiming He, Xiangyu Zhang, Shaoqing Ren, and Jian Sun. Deep residual learning for image recognition. pages 770–778, 06 2016. 

Olaf Ronneberger, Philipp Fischer, and Thomas Brox. U-net: Convolutional networks for biomedical image segmentation. In Medical Image Computing and Computer-Assisted Intervention–MICCAI 2015: 18th International Conference, Munich, Germany, October 5-9, 2015, Proceedings, Part III 18, pages 234–241. Springer, 2015. 

Kai Zhang, Wangmeng Zuo, Yunjin Chen, Deyu Meng, and Lei Zhang. Beyond a gaussian denoiser: Residual learning of deep cnn for image denoising. IEEE transactions on image processing, 26(7):3142–3155, 2017.

Weixia Zhang, Kede Ma, Guangtao Zhai, and Xiaokang Yang. Uncertainty-aware blind image quality assessment in the laboratory and wild. IEEE Transactions on Image Processing, 30:3474–3486, 2021.

Zhao Zhang, Huan Zheng, Richang Hong, Mingliang Xu, Shuicheng Yan, and Meng Wang. Deep color consistent network for low-light image enhancement. In 2022 IEEE/CVF Conference on Computer Vision and Pattern Recognition (CVPR), pages 1889–1898, 2022.

---

## [Decision Letter · Decision Letter 1]

20 Feb 2023

Semi-supervised atmospheric component learning in low-light image problem

PONE-D-23-00220R1

Dear Dr. Jung,

We’re pleased to inform you that your manuscript has been judged scientifically suitable for publication and will be formally accepted for publication once it meets all outstanding technical requirements.

Kind regards,

Praveen Kumar Donta, Ph.D.

Academic Editor

PLOS ONE

Additional Editor Comments (optional):

Reviewers' comments:

Reviewer's Responses to Questions

**Comments to the Author**

1. If the authors have adequately addressed your comments raised in a previous round of review and you feel that this manuscript is now acceptable for publication, you may indicate that here to bypass the “Comments to the Author” section, enter your conflict of interest statement in the “Confidential to Editor” section, and submit your "Accept" recommendation.

Reviewer #1: All comments have been addressed

Reviewer #3: All comments have been addressed

2. Is the manuscript technically sound, and do the data support the conclusions?

Reviewer #1: Yes

Reviewer #3: Yes

3. Has the statistical analysis been performed appropriately and rigorously? 

Reviewer #1: Yes

Reviewer #3: Yes

4. Have the authors made all data underlying the findings in their manuscript fully available?

Reviewer #1: No

Reviewer #3: Yes

5. Is the manuscript presented in an intelligible fashion and written in standard English?

Reviewer #1: Yes

Reviewer #3: Yes

6. Review Comments to the Author

Reviewer #1: (No Response)

Reviewer #3: Paper Title: Semi-supervised atmospheric component learning in low-light image problem

The paper is accepted in present form.

All the comments raised are addressed well by the authors.

7. PLOS authors have the option to publish the peer review history of their article (what does this mean?). If published, this will include your full peer review and any attached files.

Reviewer #1: No

Reviewer #3: No

---

## [Editor Report · Acceptance letter]

28 Feb 2023

PONE-D-23-00220R1 

Semi-supervised atmospheric component learning in low-light image problem 

Dear Dr. Jung:

I'm pleased to inform you that your manuscript has been deemed suitable for publication in PLOS ONE. Congratulations! Your manuscript is now with our production department. 

Kind regards, 

on behalf of

Dr. Praveen Kumar Donta 

Academic Editor

PLOS ONE